computational chemistry/organic chemistry/spectroscopy

biaryl, NMR, chemical shift, DFT, functionals, basis sets

**Author for correspondence:**
Thien T. Nguyen
e-mail: nguyentrongthien@duytan.edu.vn

This article has been edited by the Royal Society of Chemistry, including the commissioning, peer review process and editorial aspects up to the point of acceptance.

# $^1$H/$^{13}$C chemical shift calculations for biaryls: DFT approaches to geometry optimization

Thien T. Nguyen[1,2]

[1]Institute of Research and Development, and [2]Faculty of Pharmacy, College of Medicine and Pharmacy, Duy Tan University, Da Nang 550000, Vietnam

TTN, 0000-0002-2080-786X

Twelve common density functional methods and seven basis sets for geometry optimization were evaluated on the accuracy of $^1$H/$^{13}$C NMR chemical shift calculations for biaryls. For these functionals, $^1$H shifts calculations for gas phase optimized geometries were significantly less accurate than those for in-solution optimized structures, while $^{13}$C results were not strongly influenced by geometry optimization methods and solvent effects. B3LYP, B3PW91, mPW1PW91 and $\omega$B97XD were the best-performing functionals with lowest errors; among seven basis sets, DGDZVP2 and 6-31G(d,p) outperformed the others. The combination of these functionals and basis sets resulted in high accuracy with $\text{CMAE}_{min} = 0.0327$ ppm (0.76%) and 0.888 ppm (0.58%) for $^1$H and $^{13}$C, respectively. The selected functionals and basis set were validated when consistently producing optimized structures with high accuracy results for $^1$H and $^{13}$C chemical shift calculations of two other biaryls. This study highly recommends the IEFPCM/B3LYP, B3PW91, mPW1PW91 or $\omega$B97XD/DGDZVP2 or 6-31G(d,p) level of theory for the geometry optimization step, especially the solvent incorporation, which would lead to high accuracy $^1$H/$^{13}$C calculation. This work would assist in the fully structural assignments of biaryls and provide insights into in-solution biaryl conformations.

## 1. Introduction

Biaryls (figure 1) are prominent substructures found in important, chiral ligands and organocatalysts, which are useful for asymmetric transformations [1,2]. They are also present in biologically active natural products and biopolymer lignins [3,4]. Biaryls are used for sequencing lignin oligomers [5], oxidative C–C cleavage [6] and pyrolysis behaviour studies [7] and are found to be anti-oxidant and anti-inflammatory agents [8]. Owning biaryl

**Figure 1.** (*a*) Biaryls in natural products (i), ligands and catalysts (ii) and lignin structure (iii), and (*b*) Biaryls with numbering labels and the dihedral angles $\alpha$ (C1C2C2′C1′) and $\beta$ (C6C7C8C9) of biaryl **1**.

linkages, they can also be employed for biaryl conformation analysis, which is intrinsically difficult due to the absence of proton NMR spin–spin coupling information. An accurate method for predicting NMR spectra would contribute valuable insights into the conformations of biaryl structures and the local electron environment of each NMR active nucleus. Gauge-independent atomic orbital (GIAO)-DFT NMR calculations have effectively supported the structural assignments and validations of biaryl compounds with accurate predictions at affordable computational costs [9,10]. In general, the accuracy is impacted by geometry optimizations, functionals, basis sets, solvation models and NMR methods. [11–14]. For the two common nuclei of organic molecules, [1]H shift predictions are more challenging than [13]C shift ones due to significant impacts of solvation effects on protons [15].

Previous studies reported how the use of different density functional methods and basis sets for NMR calculations affected [1]H/[13]C results on a variety of organic structures [11,13,16,17]. Among those, few reports investigated the impact of optimized geometry on the NMR results. In 1997, Rablen *et al.* [18] reported the influence of molecular geometry on the calculated chemical shifts using four different levels of theory, including MP2/6-31 + G(d), B3LYP/6-31 + G(d), B3LYP/6-311 + +G(d,p) and B3LYP/6-311 + +G(2df,p) and found that such methods have small impacts on [1]H NMR calculations for small organic molecules. In 2008, Xu *et al.* [19] reported the use of functionals B3LYP, PBE0 and OPBE on 23 small, simple molecules and found that OPBE was the best-performing one. Herein, this present study shows how optimized geometries using 12 density functional methods and seven basis sets, which have been commonly employed for [1]H and [13]C NMR calculations of organic compounds, influence

the $^1$H and $^{13}$C NMR shift calculation accuracy for biaryls. Its aim is to reveal how sensitive such optimized geometries are on the chemical shift predictions of such important structure motifs and to selectively recommend density functional methods and basis sets for geometry optimization with high accuracy and an economical amount of CPU time. The broad differences of results would mean that the functional and basis set selections for geometry optimization methods are significant, while the small ones should provide confidence in carrying out such predictions.

# 2. Computational methods

Structures of **1**, **2** and **3** were initially optimized using MMFF94/MM2, which are implemented in Chem3D. The resulting geometries were used as the starting point for the individual optimizations using Gaussian09 [20], Integral equation formalism variant of the polarized continuum model (IEFPCM) was used when solvent effects were incorporated during geometry optimization [21]. Subsequent frequency calculations ensured that a potential energy surface local minimum was attained during the energy minimization. The size of integral was set at default because energy differences between optimized gas phase and in-solution structures were considered. Cartesian coordinates of such optimized geometries were given in the electronic supplementary material. For the performance evaluation of density functional methods and basis sets, local minima were employed in this investigation for economical computation. The following 12 functionals and seven basis sets were evaluated:

— Funtionals: B3LYP (Becke's three-parameter hybrid functional using B exchange and LYP correlation) [22], B3PW91 (Perdew and Wang's 1991 gradient-corrected correlation functional) [23], BPV86 (Perdew's 1986 functional) [24], CAM-B3LYP (Handy and co-workers' long-range corrected version of B3LYP using the Coulomb-attenuating method) [25], HCTH (Hamprecht-Cohen–Tozer–Handy GGA functional) [26], HSEH1PBE (The exchange part of the screened Coulomb potential of Heyd, Scuseria and Ernzerhof) [27], LSDA (Local spin-density approximation) [28], mPW1PW91 (mPW exchange and PW91 correlation) [29], PBEPBE (The functional of Perdew, Burke, and Ernzerhof) [30], TPSSTPSS (The exchange component of the Tao–Perdew–Staroverov–Scuseria) [31] and $\omega$B97XD (Head-Gordon and co-workers' dispersion corrected long-range corrected hybrid functional) [32]; M06–2X (a high-non-locality functional with double the amount of nonlocal exchange) [33].
— Basis sets: Pople's 6-31G, 6-31G(d,p), 6-31 + G(d,p), and 6-311G [34]; Dunning's cc-pVDZ correlation consistent basis set [35]; DGDZVP2 [36]; SDD (Stuttgart–Dresden effective core potential) [37].

Unless specified otherwise, single-point NMR GIAO calculations were carried out at IEFPCM (DMSO or CHCl$_3$)/$\omega$B97XD/6-31G(d,p) level of theory, which was found to consistently provide $^1$H and $^{13}$C chemical shifts with a high level of accuracy [10]. The GIAO NMR results were observed and extracted using GaussView05. Each optimized structure was used for computing the corresponding isotropic shielding constants ($\sigma_{cal}$). The chemical shifts ($\delta_{cal}$) given in the electronic supplementary material were obtained using equation (2.1). For both the $^1$H and $^{13}$C NMR calculations, an average of values of equivalents atoms was assumed. For example, a single proton/carbon signal is observed for the two methoxy groups of biaryl **1** due to fast rotations of biaryl linkage relative to the NMR measurement time scale. To reduce the systematic error of the calculations, the linear regression analysis of the calculated chemical shifts versus the experimental ones ($\delta_{exp}$) (equation (2.2)) were performed, and the scaled chemical shifts ($\delta_{scal}$) were computed according to equation (2.3). As the reference had a negligible impact on the linear regression analysis, the fixed values of 197 ppm and 31 ppm were chosen as TMS shielding constants for $^{13}$C and for $^1$H, respectively. Computed results were evaluated using mean absolute value ($|\Delta\delta|$/ppm, equation (2.4)), corrected mean absolute error (CMAE/ppm, equation (2.5)), corrected root mean squared error (CRMSE/ppm, equation (2.6) and the Pearson correlation coefficient ($r^2$). The smaller values of CMAE and CRMSE indicate smaller errors and the larger value of $r^2$ means a stronger correlation between theoretical and experimental data. Error calculations and linear correlations were performed using Microsoft Excel 2013.

$$\delta_{cal} = \sigma_{TMS} - \sigma_{cal}, \tag{2.1}$$

$$\delta_{cal} = a\delta_{exp} + b, \tag{2.2}$$

$$\delta_{scal} = \frac{\delta_{cal} - b}{a}, \tag{2.3}$$

$$|\Delta\delta| = |\delta_{scal} - \delta_{exp}|, \tag{2.4}$$

$$CMAE = \sum_{1}^{n} |\delta_{scal} - \delta_{exp}|/n \qquad (2.5)$$

and
$$CRMSE = \sqrt{\sum_{1}^{n} (\delta_{scal} - \delta_{exp})^2/n}. \qquad (2.6)$$

Figure 1b shows the numbered biaryls used for the proton and carbon atoms in this study. Due to the axial symmetry of biaryls **1**, **2** and **3**, only one side of the structures was labelled. The experimental [1]H and [13]C NMR spectra of **1** [10,38], **2** [39,40] and **3** [41,42] were reported. These compounds contain phenolic and carboxylic protons which typically did not appear in the NMR spectra due to rapid exchanges in DMSO-$d_6$ or CDCl$_3$ solvent. Therefore, this work excluded these protons in the calculations. This also excluded the main interacting atoms of the molecules leading to basis set superposition errors (BSSE). In addition, the BSSE should have a similar contribution to each NMR calculation on the same molecule. Therefore, these errors were insignificant for the comparison of performances between density functional methods and basis sets in this study.

# 3. Results and discussion

## 3.1. Functional impact

Initially, optimized geometries of biaryl 1 in gas phase and in DMSO using 12 functionals coupled with 6-31G(d,p) basis set were tested on compound 1, which was previously prepared by a two-stepped synthesis from ferulic acid [10]. The optimized structures were obtained with energies and different dihedral angles $\alpha$ (C1C2C2'C1') and $\beta$ (C6C7C8C9) (figure 2). Because the energy differences between gas phase and in-solution geometries were considered, the fine grid was used during the optimization step. Aliphatic angle $\beta$ was not altered much, while biaryl angle $\alpha$ showed a significant discrepancy when HCTH (table 1, entry 5) and LSDA (table 1, entry 7) were employed. In-solution structures were about 19 kcal/mol more stable than gas phase ones. These structures were subjected to the [1]H and [13]C chemical shift calculations and the results are shown in table 2 and figure 2.

The [1]H NMR calculations of these geometries were calculated at the level of IEFPCM(DMSO)/$\omega$B97XD/ 6-31G(d,p) [10], and their statistical parameters and mean absolute values are shown in table 2 and figure 2, respectively. Compared to other functionals, HCTH (Entry 5) and LSDA (Entry 7) showed discrepancy results with highest errors. Therefore, we excluded these two functionals in figure 2 and further discussions. These results were expected as the optimized gas phase and in-solution geometries obtained from these functionals were too different from other functionals (table 1). For other functionals, the statistical parameters for gas phase structures (0.0711 ppm (1.66%) ≤ CMAEs ≤ 0.0991 ppm (2.31%), 0.0770 ppm (1.79%) ≤ CRMSEs ≤ 0.113 ppm (2.63%) and 0.9963 ≤ $r^2$ ≤ 0.9983) were significantly less accurate than those for in-solution structures (0.0327 ppm (0.76%) ≤ CMAE ≤ 0.0549 ppm (1.28%), 0.0412 ppm (0.96%) ≤ CRMSE ≤ 0.0585 ppm (1.36%), 0.9990 ≤ $r^2$ ≤ 0.9995). This could be explained by the high exposure of hydrogen atoms to solvent molecules. The solvent effects can be clearly observed in figure 2; except for calculated **H10** shifts of the methoxy group, the mean absolute values of aromatic protons **H1** and **H5**, and methylene protons **H7** and **H8** shifts were significantly decreased when going from gas phase to in-solution structures. For in-solution geometries, noticeable deviations were consistently observed for methylene protons **H7** and **H8** ($|\Delta\delta|$ max = 0.0764 ppm, $\omega$B97XD) were observed with the noticeable deviations ranging from 0.0715 to 0.166 ppm. Therefore, solvent incorporation during the optimization process is essential for achieving the high accuracy of [1]H calculations. The three best performers, those with lowest errors, were B3PW91 (CRMSE = 0.0412 ppm, 0.96%), mPW1PW91 (CRMSE = 0.0432 ppm, 1.01%) and B3LYP (CRMSE = 0.0437 ppm, 1.02%).

[13]C results for structures optimized in gas phase and in DMSO were in high agreement with the experimental data for all tested functionals (table 3). CMAE and CRMSE values of all 12 tested functionals were in the narrow ranges of 1.09 (0.71%) to 1.50 ppm (0.98%) and 1.30 (0.85%) to 1.83 ppm (1.19%), respectively. The [13]C results were obtained with high coefficients of determination (0.9985 ≤ $r^2$ ≤ 0.9992). Compared to the [1]H results, [13]C chemical shift calculations were not strongly influenced by molecular geometries and solvation effects. This could be explained by the fact that carbon nuclei are less exposed to solvent molecules than protons. Optimized geometries using B3LYP, B3PW91, BPV86, CAM-B3LYP, HSEH1PBE, LSDA, M06-2X, mPW1PW91 and PBEPBE produced high correlated results. Among those, B3LYP is economical from the view of computational cost. The

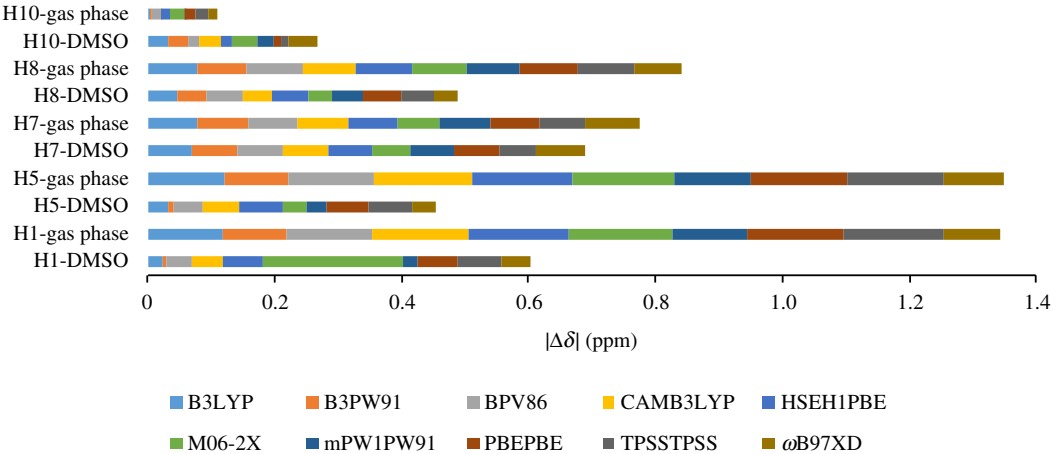

**Figure 2.** Mean absolute values of $^1H$ chemical shifts for geometries optimized in gas phase and in DMSO using 12 functionals coupled with 6-31G(d,p).

**Table 1.** Dihedral angles $\alpha$ and $\beta$ and the energy differences of optimized geometries ($\Delta E = E_{gas\ phase} - E_{in\text{-}solution}$).

| entry | functional | gas phase | | in-solution | | |
| | | $\alpha$ | $\beta$ | $\alpha$ | $\beta$ | $\Delta E$ (kcal/mol) |
|---|---|---|---|---|---|---|
| 1 | B3LYP | −132.89 | −179.06 | −128.07 | −179.76 | 19.07 |
| 2 | B3PW91 | −131.98 | −178.94 | −126.84 | −179.62 | 19.80 |
| 3 | BPV86 | −134.82 | −178.99 | −130.24 | −179.09 | 18.46 |
| 4 | CAM-B3LYP | −134.07 | −179.15 | −129.03 | −179.94 | 19.58 |
| 5 | HCTH | −125.29 | −178.93 | −114.71 | −179.92 | 18.43 |
| 6 | HSEH1PBE | −134.85 | −179.24 | −130.42 | −179.51 | 19.73 |
| 7 | LSDA | −150.22 | −173.91 | −146.66 | −174.71 | 18.66 |
| 8 | M06-2X | −134.48 | −179.42 | −130.62 | −179.79 | 19.95 |
| 9 | mPW1PW91 | −132.88 | −179.14 | −127.97 | −179.7 | 19.97 |
| 10 | PBEPBE | −135.51 | −179.29 | −131.39 | −179.14 | 18.19 |
| 11 | TPSSTPSS | −135.88 | −179.16 | −131.33 | −179.57 | 18.53 |
| 12 | $\omega$B97XD | −131.06 | −179.68 | −123.05 | −179.39 | 20.21 |

noticeable deviations of carbon atoms **C8** and **C10**, which are $sp^3$-hybridized centres and among the most shielded carbons, were consistently observed in the range of 1.58 to 2.67 ppm and 2.11 to 3.87 ppm, respectively (figure 3).

In addition, when considering only $sp^2$-hybridized carbons for the NMR calculations with a much narrower spectral window of $^{13}C$ chemical shifts (63.1 ppm), slight decreases in correlations ($0.9943 \leq r^2 \leq 0.9984$) and CRMSE values (0.94 ppm (1.49%) $\leq$ CRMSEs $\leq$ 1.44 ppm (2.28%)) for all tested density functional methods were consistently observed (figure 4). These lower correlations were expected due to the narrower spectral window. The CRMSE values were smaller in all cases but their relative percentages (CRMSE values over the spectral window) were much higher. Even though the statistical parameters for all carbon of the molecule were more meaningful than those for a carbon type due to the intrinsically electronic environments of a specific skeleton leading to characteristic magnetic properties of the active NMR carbon nucleus, the results would clarify the impact of narrowing the spectral window on calculation accuracy.

## 3.2. Basis set effects

Seven common basis sets, including 6-31G, 6-31G(d,p), 6-31G + (d,p), 6-311G, cc-pVDZ, DGDZVP2 and SDD, were coupled with B3LYP for in-solution geometry optimization steps, and NMR calculations of

**Table 2.** Accuracy evaluation of $^1H$ chemical shift calculations for optimized geometries of biaryl 1 in gas phase and in DMSO using 12 functionals coupled with 6-31G(d,p) basis set.

| entry | functional | $\delta(^1H)$-gas phase | | | $\delta(^1H)$-DMSO | | |
|---|---|---|---|---|---|---|---|
| | | $r^2$ | CMAE | CRMSE | $r^2$ | CMAE | CRMSE |
| 1 | B3LYP | 0.9976 | 0.0794 | 0.0901 | *0.9994* | *0.0406* | *0.0437*[a] |
| 2 | B3PW91 | 0.9981 | 0.0729 | 0.081 | *0.9995* | *0.0327* | *0.0412*[a] |
| 3 | BPV86 | 0.9970 | 0.0898 | 0.101 | 0.9993 | 0.0465 | 0.0499 |
| 4 | CAM-B3LYP | 0.9965 | 0.0940 | 0.110 | 0.9992 | 0.0519 | 0.0535 |
| 5 | HCTH | 0.9990 | 0.0448 | 0.058 | 0.9968 | 0.101 | 0.105 |
| 6 | HSEH1PBE | 0.9963 | 0.0989 | 0.113 | 0.9990 | 0.0549 | 0.0584 |
| 7 | LSDA | 0.9670 | 0.288 | 0.341 | 0.9820 | 0.205 | 0.250 |
| 8 | mPW1PW91 | 0.9975 | 0.0808 | 0.0917 | *0.9995* | *0.0399* | *0.0432*[a] |
| 9 | M06-2X | 0.9962 | 0.100 | 0.114 | 0.9989 | 0.0790 | 0.106 |
| 10 | PBEPBE | 0.9965 | 0.0975 | 0.110 | 0.9990 | 0.0543 | 0.0585 |
| 11 | TPSSTPSS | 0.9964 | 0.0991 | 0.112 | 0.9991 | 0.0543 | 0.0585 |
| 12 | $\omega$B97XD | 0.9983 | 0.0711 | 0.077 | 0.9992 | 0.049 | 0.0510 |

[a]The data of three best functionals are in italics.

**Table 3.** Accuracy evaluation of $^{13}C$ chemical shift calculations for optimized geometries of biaryl 1 in gas phase and in DMSO using 12 functionals coupled with 6-31G(d,p) basis set.

| entry | functional | $\delta(^{13}C)$-gas phase | | | $\delta(^{13}C)$-DMSO | | |
|---|---|---|---|---|---|---|---|
| | | $r^2$ | CMAE | CRMSE | $r^2$ | CMAE | CRMSE |
| 1 | B3LYP | 0.9992 | 1.09 | 1.34 | 0.9992 | 1.09 | 1.36 |
| 2 | B3PW91 | 0.9989 | 1.19 | 1.57 | 0.9991 | 1.13 | 1.40 |
| 3 | BPV86 | 0.9990 | 1.12 | 1.46 | 0.9992 | 1.18 | 1.30 |
| 4 | CAM-B3LYP | 0.9990 | 1.16 | 1.50 | 0.9992 | 1.09 | 1.34 |
| 5 | HCTH | 0.9986 | 1.36 | 1.75 | 0.9988 | 1.35 | 1.62 |
| 6 | HSEH1PBE | 0.9989 | 1.19 | 1.55 | 0.9992 | 1.09 | 1.35 |
| 7 | LSDA | 0.9990 | 1.18 | 1.51 | 0.9990 | 1.35 | 1.50 |
| 8 | M06-2X | 0.9988 | 1.27 | 1.63 | 0.9991 | 1.18 | 1.42 |
| 9 | mPW1PW91 | 0.9989 | 1.23 | 1.59 | 0.9991 | 1.16 | 1.42 |
| 10 | PBEPBE | 0.9990 | 1.15 | 1.48 | 0.9992 | 1.20 | 1.33 |
| 11 | TPSSTPSS | 0.9991 | 1.10 | 1.41 | 0.9985 | 1.50 | 1.83 |
| 12 | $\omega$B97XD | 0.9987 | 1.31 | 1.68 | 0.9989 | 1.27 | 1.53 |

optimized structures were carried out. Table 4 displays the dihedral angels $\boldsymbol{\alpha}$ and $\boldsymbol{\beta}$, and table 5 shows the statistical parameters. While adding polarity functions for heavy atoms and hydrogen atoms (table 5, entry 1 and 2) significantly increased the calculation accuracy, a set of the diffusion function to Pople's basis sets (table 5, entry 2 and 3) created opposite effects and cost much more computation time. 6-31G + (d,p) (table 5, entry 3), generating the most discrepancy geometries, provided $^1H$ results with low correlations and large errors. Other basis sets produced highly correlated results with CMAEs ≤ 0.0957 ppm, CMRSEs ≤ 0.111 ppm and $r^2$ ≥ 0.9987 for $^1H$ and CMAEs ≤ 1.23 ppm, CMRSEs ≤ 1.62 ppm, and $r^2$ ≥ 0.9988 for $^{13}C$. These results would allow a meaningful prediction of $^1H$ and $^{13}C$ chemical shifts. The best-performing basis set was DGDZVP2 (CMAE = 0.0361 ppm for $^1H$; CMAE = 0.888 ppm for $^{13}C$). In terms of absolute deviations, the totals of absolute deviations for protons ($\sum|\delta|$

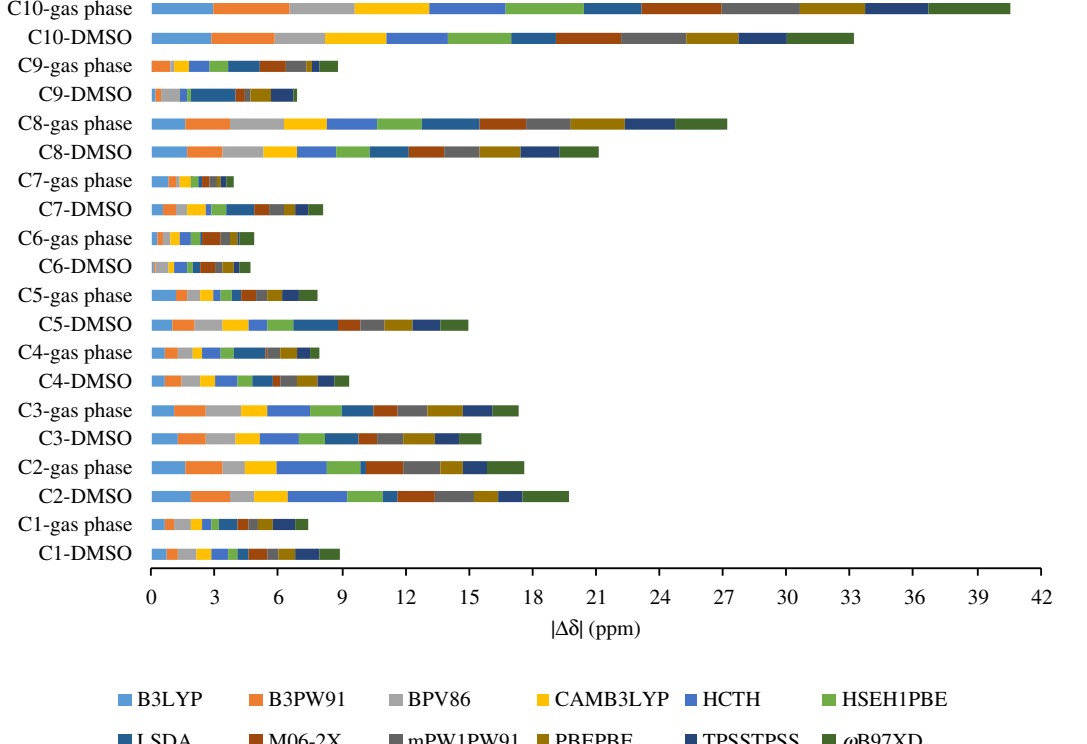

**Figure 3.** Mean absolute values of $^{13}$C chemical shifts for geometries optimized in gas phase and in DMSO using 12 functionals coupled with 6-31G(d,p).

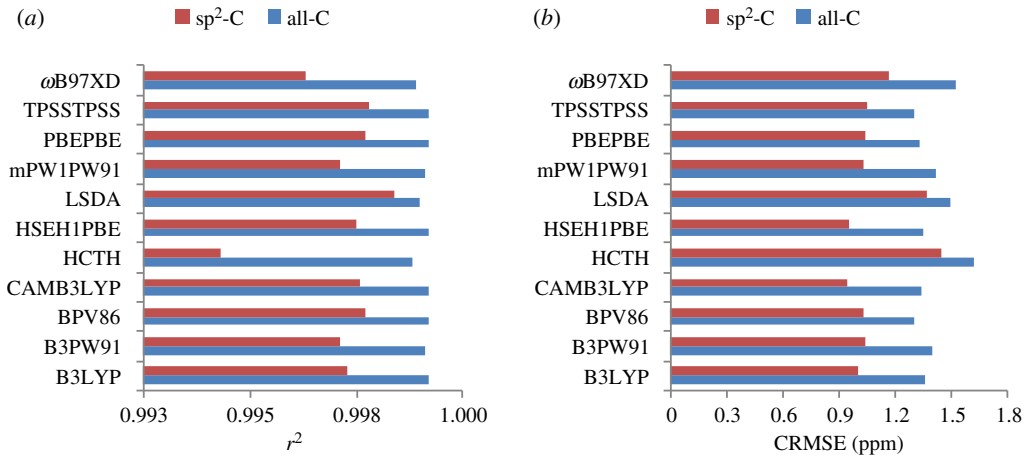

**Figure 4.** $r^2$ Values (a) and CRMSE (b) of the $^{13}$C chemical shift calculations for all carbons (All-C) and sp$^2$-hybridized ones (sp$^2$-C).

were evenly distributed while those for **C8**, **C9** and **C10** were relatively large. DGDZVP2 was more effective than the other basis sets, except for methoxy protons **H10,** aromatic carbon **C2** and methoxy carbon **C10** (figure 5).

The $^1$H/$^{13}$C calculations using the other combination of functionals and basis sets were carried out in order to assert the best-performing functionals and basis set. LSDA and HCTH functionals were excluded from these calculations due to their relatively low performances. As one of the Pople' basis sets were tested for all tested functionals (tables 2 and 3), the remaining sets would provide similar results. Therefore, cc-pVDZ, DGDZVP and SDD were coupled with 10 functionals for the calculations, and the $r^2$ values are summary in table 6 (full statistical parameters Are given in the electronic supplementary material). When using the same basis sets, $^{13}$C results were relatively less affected by the choice of functionals. The $r^2$ values were ranged from 0.9990 to 0.9994, 0.9991 to 0.9995 and 0.9980 to 0.9988 for cc-pVDZ, DGDZVP and SDD, respectively (table 6). $^1$H results showed strong

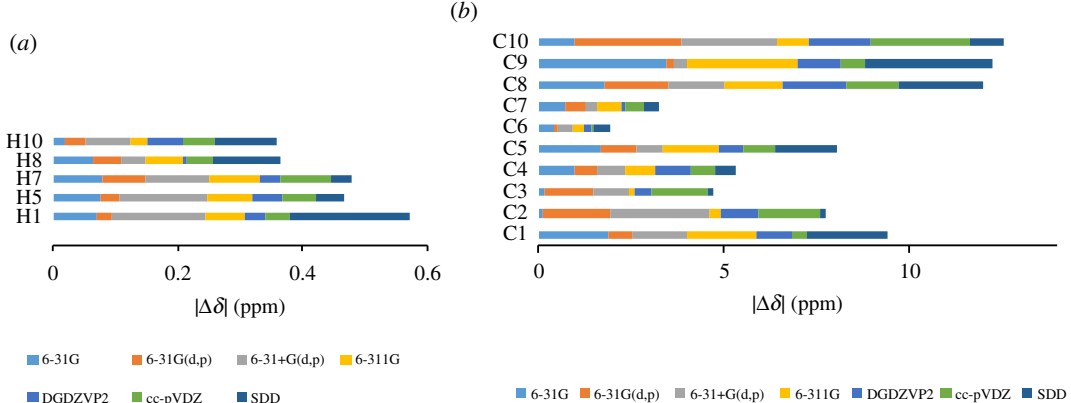

**Figure 5.** Mean absolute values of $^1$H (*a*) and $^{13}$C (*b*) chemical shifts for geometries optimized in DMSO using seven basis sets coupled with B3LYP.

**Table 4.** Dihedral angles $\alpha$ and $\beta$ of optimized geometries using seven basis sets.

| entry | basis set | $\alpha$ | $\beta$ |
|---|---|---|---|
| 1 | 6-31G | −130.29 | −179.62 |
| 2 | 6-31G(d,p) | −132.89 | −179.06 |
| 3 | 6-31G + (d,p) | −111.62 | −179.88 |
| 4 | 6-311G | −129.63 | −179.90 |
| 5 | cc-pVDZ | −129.28 | −178.97 |
| 6 | DGDZVP2 | −124.66 | −179.69 |
| 7 | SDD | −130.77 | −179.55 |

**Table 5.** Accuracy evaluation of $^{13}$C and $^1$H chemical shift calculations for optimized geometries in DMSO using seven basis sets coupled with B3LYP functional.

| entry | basis set | $\delta(^1$H) | | | $\delta(^{13}$C) | | |
|---|---|---|---|---|---|---|---|
| | | $r^2$ | CMAE | CRMSE | $r^2$ | CMAE | CRMSE |
| 1 | 6-31G | 0.9988 | 0.0616 | 0.0652 | 0.9989 | 1.21 | 1.54 |
| 2 | 6-31G(d,p) | 0.9994 | 0.0406 | 0.0437 | 0.9992 | 1.09 | 1.36 |
| 3 | 6-31G + (d,p) | 0.9965 | 0.101 | 0.109 | 0.9991 | 1.17 | 1.44 |
| 4 | 6-311G | 0.9988 | 0.0611 | 0.064 | 0.9991 | 1.10 | 1.38 |
| 5 | cc-pVDZ | 0.9991 | 0.0525 | 0.0543 | 0.9993 | 1.03 | 1.27 |
| 6 | *DGDZVP2*[a] | *0.9995* | *0.0361* | *0.0404* | *0.9995* | *0.888* | *1.03* |
| 7 | SDD | 0.9988 | 0.0957 | 0.111 | 0.9988 | 1.22 | 1.62 |

[a]The best-performing basis set and its data are in italics.

dependence on the choices of functionals and basis sets. For instance, the $r^2$ value of B3LYP/DGDZVP was 0.9995, while that of M06-2X/SDD was 0.9844. In addition to show good results obtained from $\omega$B97XD (Entry 9), the calculations confirmed that B3LYP, B3PW91 and mPW1PW91 functionals and DGDZVP basis set were consistently provided high accuracy results.

## 3.3. The use of recommended methods for biaryls 2 and 3

For validation of optimization methods, selected funtionals B3LYP, B3PW91 and mPW1PW91 and basis set DGDZVP2 were employed for optimizing biaryls **2** and **3**. The biaryl conformations of optimized

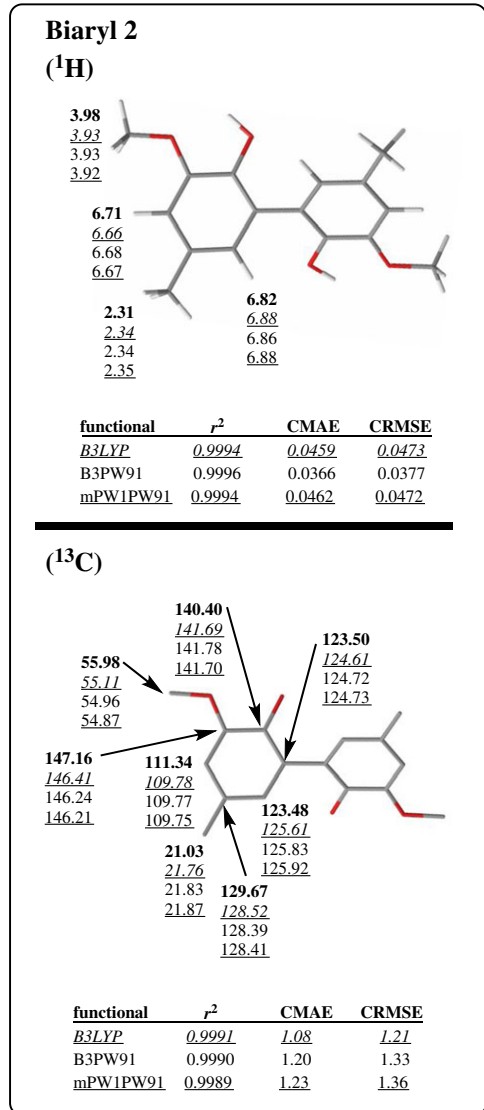

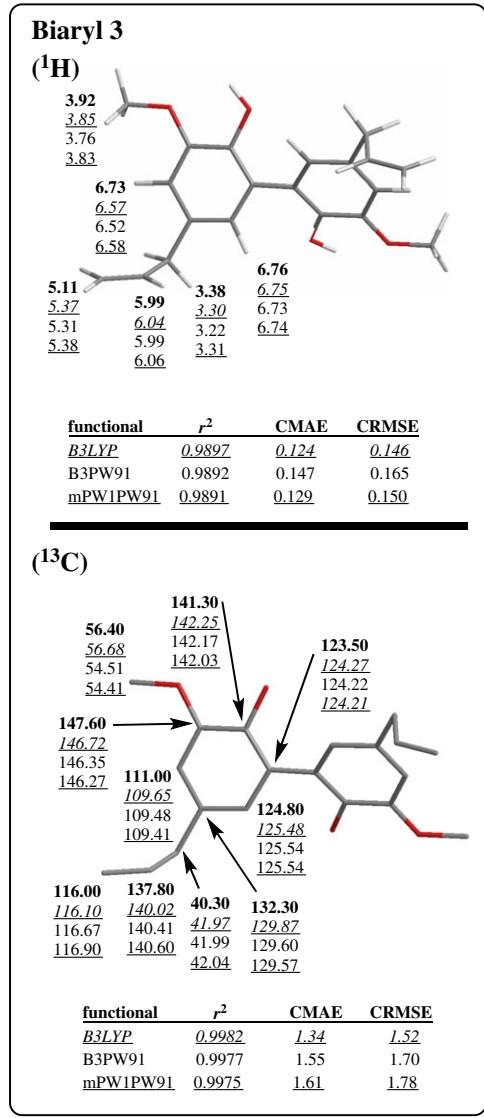

**Figure 6.** Two biaryls **2** and **3** with chemical shift data and statistical parameters for geometries optimized at IEFPCM/B3LYP/ DGDZVP2 displayed in _underlined italics_, IEFPCM/B3PW91/DGDZVP2 displayed in normal text, IEFPCM/mPW1PW91/DGDZVP2 displayed in underlined normal text, and experimental chemical shifts displayed in **bold**.

**Table 6.** $^1$H/$^{13}$C Results using the combination of 10 functionals and three basis sets. Highest accurate data for $^1$H results were in italics.

| | | cc-pVDZ | | DGDZVP | | SDD | |
|---|---|---|---|---|---|---|---|
| entry | functional | $r^2$ ($^1$H) | $r^2$ ($^{13}$C) | $r^2$ ($^1$H) | $r^2$ ($^{13}$C) | $r^2$ ($^1$H) | $r^2$ ($^{13}$C) |
| 1[a] | B3LYP | _0.9994_ | 0.9992 | _0.9995_ | 0.9995 | _0.9988_ | 0.9988 |
| 2 | B3PW91 | _0.9932_ | 0.9992 | _0.9975_ | 0.9992 | _0.9908_ | 0.9988 |
| 3 | BPV86 | 0.9848 | 0.9993 | 0.9943 | 0.9992 | 0.9892 | 0.9981 |
| 4 | CAM-B3LYP | 0.9894 | 0.9993 | 0.9945 | 0.9993 | 0.9881 | 0.9988 |
| 5 | HSEH1PBE | 0.9898 | 0.9993 | 0.9941 | 0.9993 | 0.9875 | 0.9988 |
| 6 | mPW1PW91 | _0.9922_ | 0.9992 | _0.9959_ | 0.9992 | _0.9898_ | 0.9988 |
| 7 | PBEPBE | 0.9890 | 0.9993 | 0.9942 | 0.9991 | 0.9857 | 0.9980 |
| 8 | TPSSTPSS | 0.9895 | 0.9994 | 0.9938 | 0.9992 | 0.9885 | 0.9980 |
| 9 | $\omega$B97XD | _0.9949_ | 0.9990 | _0.9979_ | 0.9992 | _0.9938_ | 0.9987 |
| 10 | M06-2X | 0.9718 | 0.9992 | 0.9922 | 0.9992 | 0.9844 | 0.9987 |

[a]These previous calculation results were included for comparison purposes. Highest accurate data for $^1$H results are in italics.

structures were adopted with the biaryl dihedral angle values ($\alpha$) of $-124° \pm 2°$ (electronic supplementary material, table S3). NMR calculations were performed and the chemical shift data and statistical parameters are shown in figure 6. In general, the calculated results were observed with low associated errors (CMAEs $\leq 0.147$ ppm and CRMSEs $\leq 0.165$ ppm for $^1$H; CMAEs $\leq 1.61$ ppm and CRMSEs $\leq 1.78$ ppm for $^{13}$C) and strong linear correlations ($r^2 \geq 9891$ for $^1$H and $r^2 \geq 0.9975$ for $^{13}$C). $^1$H chemical shifts of biaryl **3** were observed with higher errors than biaryl **2**. This could be a result of the broad, overlapping signals of an allyl proton NMR spectrum affecting the accurate assignments of experimental data. Large deviations were consistently observed for atoms **H9** ($\left| \Delta\delta_{\max} \right| = 0.269$ ppm), **H5** ($\left| \Delta\delta_{\max} \right| = 0.212$ ppm), **C8** ($\left| \Delta\delta_{\max} \right| = 2.80$ ppm) and **C6** ($\left| \Delta\delta_{\max} \right| = 2.44$ ppm) of biaryl **3** and **C1** ($\left| \Delta\delta_{\max} \right| = 2.44$ ppm) of biaryl **2**, which are sp$^2$-hybridized centres or are bound to these ones.

# 4. Conclusion

The evaluation of 12 different density functional methods and seven basis sets for geometry optimization on the accuracy of $^1$H and $^{13}$C chemical shift calculations for biaryls have been performed. The tested functionals showed relatively small impacts on $^1$H and $^{13}$C results in comparison to the basis sets. Solvent incorporation during the geometry optimization step was necessary for significantly improving $^1$H chemical shift calculation accuracy, while it has little effect on $^{13}$C chemical shifts. This result was expected as carbon nuclei are less exposed to solvent molecules than protons. The B3LYP, B3PW91, mPW1PW91 and $\omega$B97XD of 12 density functional methods were the most effective ones; among seven basis sets, DGDZVP2 and 6-31G(d,p) generated the optimized geometry with excellent $^1$H and $^{13}$C results. The combinations of these functionals and basis sets resulted in high accuracy with CMAE values as low as 0.0329 ppm (0.76%) and 0.888 ppm (0.58%) for $^1$H and $^{13}$C, respectively, which would allow meaningful predictions. The consistency of these functionals and basis set were validated on the NMR calculations of biaryls **2** and **3**. Computed $^1$H/$^{13}$C results were well correlated with the experimental data. Given the high degrees of accuracy achieved in the $^1$H/$^{13}$C chemical shift calculations, these results have provided insights into the conformation of the above biaryl linkage systems, which may be unavailable or not forthcoming using experimental NMR techniques. This work can also be useful for assisting in the full $^1$H/$^{13}$C NMR assignments of similar biaryls.

Data accessibility. The data are provided in the electronic supplementary material [43].

Competing interests. I declare I have no competing interests.

Funding. This project was supported by the International Foundation for Science (IFS), Stockholm, Sweden, through a grant no. 3-I-E-6576-1 to TTN.

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
