## [Peer Review File · Royal Society Open Science]

Review History

RSOS-210954.R0 (Original submission)

Review form: Reviewer 1

Is the manuscript scientifically sound in its present form?

Yes

Are the interpretations and conclusions justified by the results?

Yes

Is the language acceptable?

Yes

Do you have any ethical concerns with this paper?

No

Have you any concerns about statistical analyses in this paper?

No

Recommendation?

Accept as is

Comments to the Author(s)

This is a solid study. While the effects found are not large, they are meaningful. The data provided should be useful to others interested in the class of compounds examined.

The authors might also be interested in this paper: *Magn Reson Chem.* 2020;58:576

Review form: Reviewer 2**Is the manuscript scientifically sound in its present form?**

Yes

Are the interpretations and conclusions justified by the results?

No

Is the language acceptable?

Yes

Do you have any ethical concerns with this paper?

No

Have you any concerns about statistical analyses in this paper?

Yes

Recommendation?

Major revision is needed (please make suggestions in comments)

Comments to the Author(s)

The author describes twelve (12) common density functional methods and seven (7) basis sets for geometry optimization which were evaluated on the accuracy of ¹H/¹³C NMR chemical shift calculations for biaryls. For these functionals, ¹H shifts calculations for gas-phase optimized geometries were significantly less accurate than those for in-solution optimized structures, while ¹³C results were not strongly influenced by geometry optimization methods and solvent effects. B3LYP, B3PW91, mPW1PW91, and ω B97XD were the best performing functionals with lowest errors; among 7 basis sets, DGDZVP2 and 6-31G(d,p) outperformed the others. The paper could be published subject to several revisions as indicated below:

1. Page 1, lines 53-54: "In general, the accuracy is impacted by optimized geometries, functionals, basis sets, solvation models, and NMR methods."

Reference could be given to relevant references in the literature. See for example, *J. Chem. Theory Comput.* 2010, 6, 1580-1589; *Molecules* 2021, 26, 3477; *Magn. Reson. Chem.* 2021, 59, 108-116 (review article).

2. Page 1, lines 54-56: "For the two common nuclei of organic molecules, ¹H shift predictions are more challenging than ¹³C shift ones due to significant impacts of solvation effects on protons".
Page 2, lines 15-18: "For these functionals, ¹H shifts calculations for gas-phase optimized geometries were significantly less accurate than those for in-solution optimized structures, while ¹³C results were not strongly influenced by geometry optimization methods and solvent..."

The effect of solvent has been investigated in the literature, see for example, *Magn. Reson. Chem.* 2020, 58, 611-624 (review article) as well as the use of discrete solvation molecules, see *Magn. Reson. Chem.* 2016, 54, 28-38; *Molecules* 2019, 24, 2290. Reference could be given.

3. Page 3, lines 53-54. "Cartesian coordinates of such optimized geometries were given in the Supporting Information".

It is very useful that the author provides cartesian coordinates in the supporting information. The structures of the minimum energy conformers, however, should also be provided in the main text (see also comment below on the existence of hydrogen bond interactions).

4. Page 4, lines 28-31: "Unless specified otherwise, single-point NMR GIAO calculations were carried out at IEFPCM (DMSO or CHCl₃)/ ω B97XD/6-31G(d,p) level of theory, which was found to consistently provide ¹H and ¹³C chemical shifts with a high level of accuracy".

The graphical presentations and the relevant statistical data incorporate both olefinic and aliphatic protons and carbons. As a result, excellent linear correlations has been obtained. It could be of interest to provide graphical presentations and the respective statistical data by including only the aromatic and olefinic protons without the inclusion of any aliphatic protons and carbons.

5. Page 5, lines 35-39: "Aliphatic angle β was not altered much, while biaryl angle α showed significant discrepancy when HCTH (Table 1, entry 5) and LSDA (Table 1, entry 7) were employed. In-solution structures were about 19 kcal/mol more stable than gas-phase ones".

The aliphatic angle β and the biaryl angle α are not the only angles of importance. It would be extremely important to evaluate the existence of intramolecular hydrogen bond interactions as discussed in detail in my comment (8) below.

6. Page 7, lines 44-46: "Therefore, solvent incorporation during the optimization process is essential for achieving the high accuracy of ¹H calculations".

The authors should also provide computations by incorporating discrete molecules of solvent.

7. Page 10: "Table 6. ¹H/¹³C Results using the combination of 10 functionals and 3 basis sets". See above my comment (4).

8. Page 11: "Figure 5. Two biaryls 2 and 3 with chemical shift data and statistical parameters for geometries optimized at IEFPCM/B3LYP/DGDZVP2 displayed in underlined italics, IEFPCM/B3PW91/DGDZVP2 displayed in normal text, IEFPCM/mPW1PW91/DGDZVP2 displayed in underlined normal text, and experimental chemical shifts displayed in bold".

The cartoon-like structures should be replaced by the minimum energy DFT calculated structures. Particular emphasis should be given also on the possibility of an intramolecular hydrogen bond interaction between the OH group (3) with the methoxy group in position 4. From the references (Tetrahedron Letters 1998, 39, 943-946; Food Chem. 2010, 118, 256-265) provided by the author, it is clear that the OH chemical shifts are provided, although the resonances are relatively broad. Thus, DFT computations can provide a direct evidence of the existence of intramolecular hydrogen bond interaction. Please note that DFT structural investigation of methyl catechol showed the existence of intramolecular hydrogen bond interaction in CHCl₃ but not in DMSO solution (Org. Biomol. Chem. 2013, 11, 7400). I would, therefore, expect a similar phenomenon with the compounds (1), (2) and (3) of the present study.

9. Page 10, lines 53-56: "The r² values were ranged from 0.9990 to 0.9994, 0.9991 to 0.9995, and 0.9980 to 0.9988 for cc-pVDZ, DGDZVP, and SDD, respectively (Table 6). ¹H results showed strong dependence on the choices of functionals and basis sets".

Page 11, line 3: "...value of B3LYP/DGDZVP was 0.9995, while that of M06-2X/SDD was 0.9844.....".

See above my comment (4). Furthermore, it has been shown in the literature (J. Chem. Theory Comput. 2010, 6, 1580-1589; Molecules 2021, 26, 3477) that the M06-2X/SDD is not very successful in computational studies of chemical shifts.

Decision letter (RSOS-210954.R0)

Dear Dr Nguyen:

Title: $^1\text{H}/^{13}\text{C}$ Chemical shift calculations of biaryls: DFT approaches to geometry optimization
Manuscript ID: RSOS-210954

Thank you for submitting the above manuscript to Royal Society Open Science. On behalf of the Editors and the Royal Society of Chemistry, I am pleased to inform you that your manuscript will be accepted for publication in Royal Society Open Science subject to minor revision in accordance with the referee suggestions. Please find the reviewers' comments at the end of this email.

The reviewers and handling editors have recommended publication, but also suggest some minor revisions to your manuscript. Therefore, I invite you to respond to the comments and revise your manuscript.

Because the schedule for publication is very tight, it is a condition of publication that you submit the revised version of your manuscript before 28-Jul-2021. Please note that the revision deadline will expire at 00.00am on this date. If you do not think you will be able to meet this date please let me know immediately.

Supplementary files will be published alongside the paper on the journal website and posted on the online figshare repository (<https://figshare.com>). The heading and legend provided for each supplementary file during the submission process will be used to create the figshare page, so please ensure these are accurate and informative so that your files can be found in searches. Files

on figshare will be made available approximately one week before the accompanying article so that the supplementary material can be attributed a unique DOI.

Kind regards,
Dr Laura Smith
Publishing Editor, Journals

On behalf of the Subject Editor Professor Anthony Stace and the Associate Editor Dr Andrew Harned.

RSC Associate Editor:

Comments to the Author:

I believe this manuscript will prove useful for others in the field. I agree with Reviewer 1 that the study has been done carefully, but Reviewer 2 has identified a number of valid concerns that should be addressed in a revised manuscript. I will leave it up to the authors to decide how meaningful including discrete solvent molecules would be in these calculations (Reviewer 2, point 6).

RSC Subject Editor:

Comments to the Author:

(There are no comments.)

Reviewer comments to Author:

Reviewer: 1

Comments to the Author(s)

This is a solid study. While the effects found are not large, they are meaningful. The data provided should be useful to others interested in the class of compounds examined.

The authors might also be interested in this paper: Magn Reson Chem. 2020;58:576

Reviewer: 2

Comments to the Author(s)

The author describes twelve (12) common density functional methods and seven (7) basis sets for geometry optimization which were evaluated on the accuracy of ¹H/¹³C NMR chemical shift calculations for biaryls. For these functionals, ¹H shifts calculations for gas-phase optimized geometries were significantly less accurate than those for in-solution optimized structures, while

¹³C results were not strongly influenced by geometry optimization methods and solvent effects. B3LYP, B3PW91, mPW1PW91, and ω B97XD were the best performing functionals with lowest errors; among 7 basis sets, DGDZVP2 and 6-31G(d,p) outperformed the others. The paper could be published subject to several revisions as indicated below:

1. Page 1, lines 53-54: "In general, the accuracy is impacted by optimized geometries, functionals, basis sets, solvation models, and NMR methods."

Reference could be given to relevant references in the literature. See for example, J. Chem. Theory Comput. 2010, 6, 1580-1589; Molecules 2021, 26, 3477; Magn. Reson. Chem. 2021, 59, 108-116 (review article).

2. Page 1, lines 54-56: "For the two common nuclei of organic molecules, ¹H shift predictions are more challenging than ¹³C shift ones due to significant impacts of solvation effects on protons". Page 2, lines 15-18: "For these functionals, ¹H shifts calculations for gas-phase optimized geometries were significantly less accurate than those for in-solution optimized structures, while ¹³C results were not strongly influenced by geometry optimization methods and solvent..."

The effect of solvent has been investigated in the literature, see for example, Magn. Reson. Chem. 2020, 58, 611-624 (review article) as well as the use of discrete solvation molecules, see Magn. Reson. Chem. 2016, 54, 28-38; Molecules 2019, 24, 2290. Reference could be given.

3. Page 3, lines 53-54. "Cartesian coordinates of such optimized geometries were given in the Supporting Information".

It is very useful that the author provides cartesian coordinates in the supporting information. The structures of the minimum energy conformers, however, should also be provided in the main text (see also comment below on the existence of hydrogen bond interactions).

4. Page 4, lines 28-31: "Unless specified otherwise, single-point NMR GIAO calculations were carried out at IEFPCM (DMSO or CHCl₃)/ ω B97XD/6-31G(d,p) level of theory, which was found to consistently provide ¹H and ¹³C chemical shifts with a high level of accuracy".

The graphical presentations and the relevant statistical data incorporate both olefinic and aliphatic protons and carbons. As a result, excellent linear correlations has been obtained. It could be of interest to provide graphical presentations and the respective statistical data by including only the aromatic and olefinic protons without the inclusion of any aliphatic protons and carbons.

5. Page 5, lines 35-39: "Aliphatic angle β was not altered much, while biaryl angle α showed significant discrepancy when HCTH (Table 1, entry 5) and LSDA (Table 1, entry 7) were employed. In-solution structures were about 19 kcal/mol more stable than gas-phase ones".

The aliphatic angle β and the biaryl angle α are not the only angles of importance. It would be extremely important to evaluate the existence of intramolecular hydrogen bond interactions as discussed in detail in my comment (8) below.

6. Page 7, lines 44-46: "Therefore, solvent incorporation during the optimization process is essential for achieving the high accuracy of ¹H calculations".

The authors should also provide computations by incorporating discrete molecules of solvent.

7. Page 10: "Table 6. ¹H/¹³C Results using the combination of 10 functionals and 3 basis sets". See above my comment (4).

8. Page 11: "Figure 5. Two biaryls 2 and 3 with chemical shift data and statistical parameters for geometries optimized at IEFPCM/B3LYP/DGDZVP2 displayed in underlined italics, IEFPCM/B3PW91/DGDZVP2 displayed in normal text, IEFPCM/mPW1PW91/DGDZVP2 displayed in underlined normal text, and experimental chemical shifts displayed in bold".

The cartoon-like structures should be replaced by the minimum energy DFT calculated structures. Particular emphasis should be given also on the possibility of an intramolecular hydrogen bond interaction between the OH group (3) with the methoxy group in position 4. From the references (Tetrahedron Letters 1998, 39, 943-946; Food Chem. 2010, 118, 256-265) provided by the author, it is clear that the OH chemical shifts are provided, although the resonances are relatively broad. Thus, DFT computations can provide a direct evidence of the existence of intramolecular hydrogen bond interaction. Please note that DFT structural investigation of methyl catechol showed the existence of intramolecular hydrogen bond

interaction in CHCl₃ but not in DMSO solution (Org. Biomol. Chem. 2013, 11, 7400). I would, therefore, expect a similar phenomenon with the compounds (1), (2) and (3) of the present study. 9. Page 10, lines 53-56: "The r² values were ranged from 0.9990 to 0.9994, 0.9991 to 0.9995, and 0.9980 to 0.9988 for cc-pVDZ, DGDZVP, and SDD, respectively (Table 6). 1H results showed strong dependence on the choices of functionals and basis sets".

Page 11, line 3: "...value of B3LYP/DGDZVP was 0.9995, while that of M06-2X/SDD was 0.9844.....".

See above my comment (4). Furthermore, it has been shown in the literature (J. Chem. Theory Comput. 2010, 6, 1580-1589; Molecules 2021, 26, 3477) that the M06-2X/SDD is not very successful in computational studies of chemical shifts.

Author's Response to Decision Letter for (RSOS-210954.R0)

See Appendix A.

Decision letter (RSOS-210954.R1)

Dear Dr Nguyen:

Title: 1H/13C Chemical shift calculations for biaryls: DFT approaches to geometry optimization
Manuscript ID: RSOS-210954.R1

It is a pleasure to accept your manuscript in its current form for publication in Royal Society Open Science. The chemistry content of Royal Society Open Science is published in collaboration with the Royal Society of Chemistry.

Yours sincerely,
Dr Ellis Wilde
Publishing Editor, Journals

Royal Society of Chemistry
Thomas Graham House

Science Park, Milton Road
Cambridge, CB4 0WF
Royal Society Open Science - Chemistry Editorial Office

On behalf of the Subject Editor Professor Anthony Stace and the Associate Editor Dr Andrew Harned.

RSC Associate Editor

Comments to the Author:

The author has carefully considered and responded appropriately to the original referees. I can recommend publication at this time.

Reviewer(s)' Comments to Author:

Appendix A

Response to the comments of Editor and Reviewers.

RSC Associate Editor:

Comments to the Author:

I believe this manuscript will prove useful for others in the field. I agree with Reviewer 1 that the study has been done carefully, but Reviewer 2 has identified a number of valid concerns that should be addressed in a revised manuscript. I will leave it up to the authors to decide how meaningful including discrete solvent molecules would be in these calculations (Reviewer 2, point 6).

This author would like to thank the Editor for his/her comments. The discrete solvation model is typically effective for polar protons having inter- and/or intra-molecular hydrogen bonds. In the biaryl structures of this study, the polar protons are phenolic and carboxylic protons. However, these protons are not considered in this work for the reason that their chemical shifts are strongly depend on the polarity of solvents and the concentrations of solutions (Please see respond to comments 5 and 8 for more information). This work mainly focuses on the evaluation of density functional methods and basis sets for geometry optimizations and exclusively employs integral equation formalism polarized continuum model (IEFPCM) for the optimizations and NMR calculations. Given the high accuracy results obtained for considered protons and carbon nuclei, the IEFPCM model is proved to be suitable for the evaluation in this study. However, this author would strongly consider the comparison of continuum, discrete, and hybrid solvation models in studying ¹H chemical shift calculations for the polar protons of biaryls in the future studies.

RSC Subject Editor:

Comments to the Author:

(There are no comments.)

Reviewer comments to Author:

Reviewer: 1

Comments to the Author(s)

This is a solid study. While the effects found are not large, they are meaningful. The data provided should be useful to others interested in the class of compounds examined.

The authors might also be interested in this paper: Magn Reson Chem. 2020;58:576

The author would like to thank the Reviewer for his/her comments.

Reviewer: 2

Comments to the Author(s)

The author describes twelve (12) common density functional methods and seven (7) basis sets for geometry optimization which were evaluated on the accuracy of ¹H/¹³C NMR chemical shift calculations for biaryls. For these functionals, ¹H shifts calculations for gas-phase optimized geometries were significantly less accurate than those for in-solution optimized structures, while ¹³C results were not strongly influenced by geometry optimization methods and solvent effects. B3LYP, B3PW91, mPW1PW91, and ωB97XD were the best performing functionals with lowest errors; among 7 basis sets, DGDZVP2 and 6-31G(d,p) outperformed the others. The paper could be published subject to several revisions as indicated below:

1. Page 1, lines 53-54: "In general, the accuracy is impacted by optimized geometries, functionals, basis sets, solvation models, and NMR methods."

Reference could be given to relevant references in the literature. See for example, J. Chem. Theory Comput. 2010, 6, 1580-1589; Molecules 2021, 26, 3477; Magn. Reson. Chem. 2021, 59, 108-116 (review article).

The author would like to thank the Reviewer for his/her comments.

Respond to comment 1. The author accordingly cited J. Chem. Theory Comput. 2010, 6, 1580-1589. The other two references focusing on the accurate calculations of ¹H/¹³C chemical shifts of organic compounds are not general regarding the scope of density functional methods and basis sets. Therefore, this author did not include these references.

2. Page 1, lines 54-56: "For the two common nuclei of organic molecules, ¹H shift predictions are more challenging than ¹³C shift ones due to significant impacts of solvation effects on protons".

Page 2, lines 15-18: "For these functionals, ¹H shifts calculations for gas-phase optimized geometries were significantly less accurate than those for in-solution optimized structures, while ¹³C results were not strongly influenced by geometry optimization methods and solvent..."

The effect of solvent has been investigated in the literature, see for example, Magn. Reson. Chem. 2020, 58, 611-624 (review article) as well as the use of discrete solvation molecules, see Magn. Reson. Chem. 2016, 54, 28-38; Molecules 2019, 24, 2290. Reference could be given.

Respond to comment 2. The review article (Magn. Reson. Chem. 2020, 58, 611-624) mainly covers the ¹H, ¹³C, and ¹⁵N NMR calculations incorporating implicit and explicit solvent models. This reference was cited accordingly.

The other two research articles mainly focus on the ¹H shift calculation for phenolic and alcoholic protons having intra- and/or inter-molecular hydrogen bonds. In the current study, these types of protons are not considered (Please see respond to comments 5 and 8 for more details). Therefore, this author did not include these references.

3. Page 3, lines 53-54. "Cartesian coordinates of such optimized geometries were given in the Supporting Information".

It is very useful that the author provides cartesian coordinates in the supporting information. The

structures of the minimum energy conformers, however, should also be provided in the main text (see also comment below on the existence of hydrogen bond interactions).

Respond to comment 3. A figure of too many biaryl geometries optimized at a variety of DFT theory levels would not meaningfully improve the manuscript. Therefore, the two representative geometries of the biaryl structures showing the preferred biaryl conformations were incorporated in Figure 6 of the main text (Page 12 of the manuscript). In addition, the 3D-structures of the three biaryl were also presented in the graphical table of content (TOC).

4. Page 4, lines 28-31: "Unless specified otherwise, single-point NMR GIAO calculations were carried out at IEFPCM (DMSO or CHCl₃)/ ω B97XD/6-31G(d,p) level of theory, which was found to consistently provide ¹H and ¹³C chemical shifts with a high level of accuracy".

The graphical presentations and the relevant statistical data incorporate both olefinic and aliphatic protons and carbons. As a result, excellent linear correlations has been obtained. It could be of interest to provide graphical presentations and the respective statistical data by including only the aromatic and olefinic protons without the inclusion of any aliphatic protons and carbons.

7. Page 10: "Table 6. ¹H/¹³C Results using the combination of 10 functionals and 3 basis sets". See above my comment (4).

9. Page 10, lines 53-56: "The r² values were ranged from 0.9990 to 0.9994, 0.9991 to 0.9995, and 0.9980 to 0.9988 for cc-pVDZ, DGDZVP, and SDD, respectively (Table 6). ¹H results showed strong dependence on the choices of functionals and basis sets".

Page 11, line 3: "...value of B3LYP/DGDZVP was 0.9995, while that of M06-2X/SDD was 0.9844.....".

See above my comment (4). Furthermore, it has been shown in the literature (J. Chem. Theory Comput. 2010, 6, 1580-1589; Molecules 2021, 26, 3477) that the M06-2X/SDD is not very successful in computational studies of chemical shifts.

Respond to comments 4, 7, and 9. The author accordingly calculated r² and CRMSE values for only sp²-hybridized carbon nucleus. The correlations were slightly decreased in all cases. Figure 4 showing the changes of r² and CMAE values was added to illustrate this point (Please see page 9 of the manuscript). These lower correlations resulted from narrowing the ¹³C spectral window were expected. No comparison was made for aromatic protons because there are only two of these presented in the molecules. This author would think the statistical parameters for all carbons of the molecules are more meaningful than those for a carbon type due to the intrinsically electronic environments of a specific structure skeleton leading to the characteristic magnetic properties of its active NMR nuclei.

In addition, the evaluation of M06-2X/SDD method for geometry optimization (Table 6) is for avoiding any pseudo-potential regarding the combinations of density functional methods and basis sets.

5. Page 5, lines 35-39: "Aliphatic angle β was not altered much, while biaryl angle α showed significant discrepancy when HCTH (Table 1, entry 5) and LSDA (Table 1, entry 7) were employed. In-solution structures were about 19 kcal/mol more stable than gas-phase ones". The aliphatic angle β and the biaryl angle α are not the only angles of importance. It would be extremely important to evaluate the existence of intramolecular hydrogen bond interactions as discussed in detail in my comment (8) below.

8. Page 11: "Figure 5. Two biaryls 2 and 3 with chemical shift data and statistical parameters for geometries optimized at IEFPCMB3LYP/DGDZVP2 displayed in underlined italics, IEFPCMB3PW91/DGDZVP2 displayed in normal text, IEFPCM/mPW1PW91/DGDZVP2 displayed in underlined normal text, and experimental chemical shifts displayed in bold". The cartoon-like structures should be replaced by the minimum energy DFT calculated structures. Particular emphasis should be given also on the possibility of an intramolecular hydrogen bond interaction between the OH group (3) with the methoxy group in position 4. From the references (Tetrahedron Letters 1998, 39, 943-946; Food Chem. 2010, 118, 256-265) provided by the author, it is clear that the OH chemical shifts are provided, although the resonances are relatively broad. Thus, DFT computations can provide a direct evidence of the existence of intramolecular hydrogen bond interaction. Please note that DFT structural investigation of methyl catechol showed the existence of intramolecular hydrogen bond interaction in CHCl₃ but not in DMSO solution (Org. Biomol. Chem. 2013, 11, 7400). I would, therefore, expect a similar phenomenon with the compounds (1), (2) and (3) of the present study.

Respond to comments 5 and 8. The cartoon-like structures were replaced by the two optimized geometries of biaryls 2 and 3. Even though phenolic protons having inter- and/or intra-molecular hydrogen bonds are highly interested, these protons are beyond the scope of this study. The chemical shifts of these protons would strongly depend on the polarity of solvents and the concentration of the solution. This author would exclusively study these chemical shifts using the combination of both experimental and computational NMR methods in the future. Besides the density functional methods and basis sets, the investigation would also include the impact of concentrations and solvents on the chemical shifts and the use of implicit, explicit, and hybrid solvation models so that high accuracy results can be achieved. This author is highly appreciated the Reviewer for providing the reference (Org. Biomol. Chem. 2013, 11, 7400) investigating the phenolic protons of phenol, 4-methylcatechol and the natural product genkwanin. This reference will be very useful for our future studies.

6. Page 7, lines 44-46: "Therefore, solvent incorporation during the optimization process is essential for achieving the high accuracy of 1H calculations". The authors should also provide computations by incorporating discrete molecules of solvent.

Respond to comment 6. Please see the above respond to RSC associate editor.